# Preventing Laboratory-Acquired Brucellosis in the Era of MALDI-TOF Technology and Molecular Tests: A Narrative Review

**Pablo Yagupsky**

Clinical Microbiology Laboratory, Soroka Medical Center, Ben-Gurion University of the Negev, Beer-Sheva 84101, Israel; pyagupsky@gmail.com; Tel.: +972-506264359

**Simple Summary:** Brucellosis is a bacterial zoonosis transmitted to humans by exposure to infected animals and their products. Although human brucellosis is rarely fatal, the disease is difficult to diagnose, mimics other infectious and non-infectious conditions, requires prolonged antibiotic therapy, and may lead to chronicity and complications. Brucellae are highly contagious to man because of their low infecting dose and multiple routes of transmission, including, among others, the gastrointestinal and respiratory tracts and abraded skin. The culture diagnosis of the disease is particularly risky, requiring the use of Class II biological safety cabinets to avoid exposure and prevent laboratory-acquired infections. In recent years, culture-independent molecular tests have substantially improved the diagnosis of the disease, while brucellar identification by MALDI-TOF technology has reduced the dangerous handling of the organism. Unfortunately, these novel and safer methods are costly and, thus, frequently unavailable in developing countries where brucellosis is endemic.

**Abstract:** Brucellosis is one of the most common etiologies of laboratory-acquired infections worldwide, and handling of living brucellae should be performed in a Class II biological safety cabinet. The low infecting dose, multiple portals of entry to the body, the wide variety of potentially contaminated specimens, and the unspecific clinical manifestations of human infections facilitate the unintentional transmission of brucellae to laboratory personnel. Work accidents such as spillage of culture media cause only a small minority of exposures, whereas >80% of events result from unfamiliarity with the phenotypic features of the genus, misidentification of isolates, and unsafe laboratory practices such as working on an open bench without protective goggles or gloves or the aerosolization of bacteria. The bacteriological diagnosis of brucellae by traditional methods is simple and straightforward but requires extensive manipulation of the isolates, and, nowadays, many laboratory technicians are not familiar with the genotypic features of the genus, resulting in inadvertent exposure and contagion. Detection of brucellar infections by culture-independent molecular methods is safe, but the identification of the organism using MALDI-TOF technology is not hazard-free, requiring an initial bacterial inactivation step to avoid transmission. Unfortunately, these novel and safer methods are costly and frequently unavailable in resource-limited endemic countries.

**Keywords:** laboratory-acquired brucellosis; prevention; cultures; identification; molecular methods; MALDI-TOF

## 1. Introduction

In 2019 the inadvertent release of aerosols contaminated with the *Brucella suis* S2 vaccine strain from the fermentation tanks of a biopharmaceutical plant located in Lanzhou, Northwestern China, resulted in a massive outbreak of laboratory-acquired brucellosis (LAB) [1]. The outbreak involved factory personnel and spread to neighboring communities, affecting >10,000 residents [1]. The event represents a tragic and tangible reminder of the high transmissibility of members of the genus and the potential role of brucellae as

bioterrorism agents [2]. Although the Lanzhou outbreak stands out because of its enormous size, it should be pointed out that brucellosis is one of the most common organisms transmitted in the laboratory setting, and smaller LAB clusters have repeatedly occurred worldwide [3,4].

In the last two decades, molecular detection tests that do not require the isolation of dangerous *Brucella* organisms [5–7] and matrix-assisted laser desorption ionization-time of flight mass spectrometry (MALDI-TOF) identification technology that substantially reduced the manipulation of living bacteria [7–9] have been introduced into clinical practice. Although less hazardous than traditional bacteriological methods, the novel approaches are not risk-free and present new biosafety challenges. The present narrative review summarizes the factors involved in the causation of laboratory-acquired brucellar infections, the prevention of occupational exposures to the organism, and their management, with special emphasis on the biosafety implications of using the new detection and identification methods.

## 2. *Brucella*: A Highly Transmissible Organism

Brucellae are small Gram-negative facultative intracellular coccobacilli that infect a variety of feral and domestic animals [10,11]. The genus currently comprises at least 12 recognized species, of which 4, namely *B. melitensis*, *B. abortus*, *B. suis*, and *B. canis,* are the main etiologic agents of human disease [11]. Although each one of these species is associated with preferential animal hosts (*B. melitensis* with small ruminants, *B. abortus* with cattle, *B. suis* with swine, and *B. canis* with canids), brucellae can be transmitted to non-canonical animals, including humans [10,11]. In most cases, human infection results from intimate contact with diseased animals or consumption of contaminated dairy products, whereas person-to-person contagion is exceptional [12]. In the past, this zoonosis had a worldwide distribution, but the implementation of rigorous control policies in industrialized countries, consisting of periodic screening of livestock, culling of infected animals, and routine vaccination of herds, have effectively controlled the disease. In industrialized countries, human brucellosis is, thus, uncommon, and most cases can be traced to the occupational exposure of veterinarians, laboratory personnel, and abattoir workers, and foreign travels to endemic regions or illegal imports of contaminated foodstuff [11]. Brucellosis, however, remains highly prevalent in Mediterranean countries, the Middle East, Latin America, the Indian subcontinent, and Africa, where half a million new cases of human infection are detected annually [11].

*Brucella* species are characterized by several biological features that facilitate their easy transmission to laboratory personnel (Table 1), making the organism one of the most common etiologies of laboratory-acquired infections: the infecting dose of aerosolized bacteria is low, ranging from 10 and 100 organisms. Brucellae may penetrate the human body through portals of entry that are relevant to the laboratory work and, especially, the respiratory tract and conjunctival epithelium, but also abraded and uncovered skin and the gastrointestinal tract. The attack rate in the Clinical Microbiology Laboratory (CML) setting is high, ranging between 30% and 100%, depending on the inoculum, the physical location of the workers, and the source of the exposure [13–15]. It grows on routine culture media such as blood- and chocolate-agar, and colonies exhibit an indistinctive appearance. Although the organism does not produce spores, it may persist on inanimate surfaces for weeks and even months [16].

**Table 1.** Hazards and factors involved in laboratory-acquired brucellosis.

| Category | Hazard | |
|---|---|---|
| **Bacteriological features of brucellae** | Low infecting dose | |
| | Multiple portals of entry to the human body | |
| | High infectivity | |
| | Long-term persistence on inanimate surfaces | |
| | Exponential biomass growth during incubation | |
| **Epidemiology** | High burden of disease in endemic areas | |
| **Clinical disease** | Unspecific symptoms and signs | |
| | Mimics other infectious and non-infectious conditions | |
| | Lack of communication with the laboratory | |
| | Contamination of a wide diversity of clinical specimens | |
| **Identification of the isolate** | Unfamiliarity with the genus in non-endemic regions | |
| | Inconspicuous appearance of colonies | |
| | Misleading Gram stain | |
| | Misidentification by | commercial biochemical kits |
| | | MALDI-TOF technology |
| | | molecular methods |
| **Unsafe laboratory practices** | Lack of biosafety protocols | |
| | Lack of personal protective equipment | |
| | Work in an open bench | |
| | Eating, drinking, or smoking at the workstation | |
| | Aerosolization of living bacteria by centrifugation, vortexing, catalase test, inadequate sterilization of exhaust gas, and malfunction of biological safety cabinets | |
| | Accidents such as spillage of media, breakage of tubes, and needle stick injuries | |
| **Environment and laboratory equipment** | Crowding | |
| | Poorly designed ventilation systems | |
| | Malfunction or improper use of biological safety cabinets | |

### 3. Human Brucellosis, a "Great Imitator"

Human brucellosis exhibits a wide range of clinical severity, from asymptomatic infections and a mild "flu-like disease" to life-threatening meningoencephalitis and endocarditis [11]. The disease may affect different body organs such as the joints and bones, the liver, the genital tract, and the central nervous system, mimicking other infectious and non-infectious conditions, and the true nature of the disease may not be suspected, and the diagnosis delayed or missed altogether [10,11]. Even in areas endemic to zoonosis, the diagnosis of brucellosis is not initially considered in a substantial fraction of patients [17]. Under these circumstances, the physicians may fail to alert the CML that the patients' specimens might contain a hazardous pathogen and should be handled with appropriate safety precautions [3]. The vague and unspecific manifestations of the disease may also result in a delay in recognition of outbreaks of LAB and failure to implement corrective measures and prevent additional cases [4]. Another implication of the protean manifestations of the disease is the wide variety of clinical specimens that can harbor *Brucella* organisms and be submitted to the CML. Although blood and synovial fluid aspirates are the most frequently contaminated samples, biopsy material, bone marrow, cerebrospinal fluid, urogenital specimens, placentae, and amniotic fluid may also represent unforeseen sources of occupational exposure [18].

The burden of the disease in endemic areas can be appalling, posing a permanent threat to laboratory personnel. Whereas in the United States, approximately 120 cases of brucellosis are reported annually countrywide, and LAB events are rare [4]. In a single CML in Ankara, Turkey, a mean of 400 clinical specimens yield *Brucella* organisms each year, and LAB affects 10 of 55 (18%) technicians, with an annual risk of 8% per employee [19].

The incubation of the disease in humans is highly variable, spanning from a few days to months, and 21% of LAB cases have an onset >12 weeks after exposure, implying that exposed personnel should be closely followed-up for the appearance of clinical symptoms and seroconversion for a prolonged period [4].

## 4. Diagnosing Human Brucellosis

A prompt and clear-cut diagnosis of human brucellosis is critical for the patient's management because successful antibiotic therapy requires prolonged administration of drug combinations that are not employed for other infections, and unless the organism is eradicated at the early stages of the disease, brucellosis may run a chronic and complicated clinical course [11]. Furthermore, the diagnosis of brucellosis in humans has serious public health significance because it implies contact with a zoonotic source that has to be traced, identified, and controlled, or could represent a bioweapon attack [2,20].

The laboratory confirmation of the diagnosis has traditionally relied on the cultural isolation and identification of the agent by biochemical means and/or positive serological tests. The isolation of *Brucella* species remains a suboptimal diagnostic tool, and the sensitivity of the culture is substantially reduced in protracted and/or focal infections [21]. In recent years, novel culture-independent nucleic acid amplification tests (NAATs) have been added to the diagnostic armamentarium, enabling the safe and rapid detection of the bacterium [7]. However, the recovery of the organism has not been abandoned, and the isolation of brucellae from blood, other normally sterile body fluids, and tissues are irrefutable proof of active infection [10]. From an epidemiological point of view, isolation enables speciation and genotyping, making it possible to track the source and discriminates between wild and vaccine strains [22]. A positive culture is also important for diagnosis at the initial stages of infection, when the results of the serological tests are still negative or show non-diagnostic or borderline antibody titers [23] and enables the performance of antibiotic susceptibility testing of the isolate when indicated. An important benefit of isolation is the fact that it establishes the diagnosis in cases in which the disease is not clinically suspected. Unsurprisingly, brucellae are unexpectedly recovered from a blood culture obtained as part of the routine workup of a febrile patient [17,24], whereas ordering a serological assay or a species-specific NAAT requires considering a priori the possibility of brucellosis. Diagnosis by serological means has the advantages of simplicity and low cost, which are especially relevant to endemic and remote rural regions where more sophisticated and expensive tools are scarce or non-existent [7]. The approach, however, has several drawbacks: it has low sensitivity in the initial stages of the infection, protracted cases, and focal infections; the specificity is limited by cross-reacting antigens of taxonomically related and unrelated bacterial species; interpretation of the serological test results may be difficult in individuals repeatedly exposed to the organism [7]. Although NAATs have an unsurpassed sensitivity and safety profile, the high cost and unavailability of sophisticated molecular technology in resource-poor endemic areas, as well as the lack of standardization and reproducibility of the different methods and commercial kits, limit their routine use [7]. Additionally, a positive NAAT cannot discriminate between active disease and past and resolved brucellar infection [7].

## 5. *Brucella* Cultures and Laboratory Safety

The concentration of viable brucellae in blood and other clinical samples is variable, ranging from 1 colony-forming unit (CFU)/mL to >1000 CFUs/mL, being higher in the early stages of the disease and decreasing over time as the result of a mounting immune response [25,26]. Whereas grinding and homogenizing tissues are risky procedures that

must be performed in a safety cabinet, normally sterile body fluids other than reproductive specimens (amniotic fluid, placental products) are not considered to represent a substantial risk of transmission unless a flagrant breach of laboratory safety practices has been committed. The contagion risk increases exponentially during and after incubation, and colonies growing on an agar plate and positive blood culture vials contain millions of living and highly transmissible bacteria. Overall, 142 of 167 (85%) laboratory workers exposed and 46 of 71 (65%) LAB cases reviewed by Traxler et al. occurred in CMLs, followed by research and reference labs and vaccine production facilities [1,4]. Laboratory accidents such as breaking of centrifuge vials [27] or blood-culture bottles [28], self-inoculation of a suspension of brucellae [29] or a patient's synovial fluid [28], and spillage of culture broths played a minor role in LAB events and caused only 18 of 165 (11%) exposures [4]. More commonly, transmission is the result of unsafe working practices, such as handling culture media on an open bench top [13,28,30–32], not using protective equipment [19], sniffing plates [14,19,33–35], or ingesting suspensions of living organisms during mouth pipetting [29]. Disregarding the portal of entry to the human body, brucellae are translocated to the regional lymph nodes and subsequently transferred to the bloodstream causing continuous bacteremia and invasion of macrophages-rich body tissues and organs, such as the bone marrow, lymph nodes, spleen, and liver, where they persist, adopting a facultative intracellular lifestyle [10]. Therefore, blood cultures are suitable specimens for detecting circulating brucellae, especially at the initial stages of the infection. Blood samples are also easy to obtain and repeat, and drawing multiple specimens increases the detection sensitivity [7]. Thus, blood samples are the most common clinical specimens from which *Brucella* species are isolated in the CML, and their handling represents the most common source of LAB.

Modern automated blood culture systems detect the presence of microorganisms by continuously monitoring rising $CO_2$ levels in the inoculated vials released by multiplying bacteria or fungi [7]. The measurement is performed without piercing the vial top and thus, no nebulization of viable bacteria occurs. However, once the $CO_2$ level reaches the positivity threshold, the broth is aspirated, subcultured on solid media and incubated, and a Gram stain is performed [7]. Bacterial colonies developing on the agar surface are then subjected to a variety of biochemical tests to enable the identification of the isolate. Bacteriological procedures such as centrifugation and vortexing of bacterial suspensions and the performance of subcultures and biochemical testing may result in dispersion and spillage of living bacteria, contamination of the laboratory environment, and unintentional transmission to the working personnel. Although *Brucella* species have been traditionally considered slow-growing bacteria [4], the Bactec blood culture system (Becton Dickinson Diagnostic Instrument Systems, Towson, MD, USA) enabled the detection of *B. melitensis* in 33 of 42 (78.6%) pediatric and 27 of 31 (87.1%) adult blood cultures from Israeli patients with acute infections within a 72 h incubation [36,37]. Therefore, a short time-to-detection does not reliably exclude the presence of brucellae in the blood culture vial.

Since *Brucella* organisms undergo phagocytosis and tend to circulate in the bloodstream inside mononuclear phagocytic cells, the Isolator Microbial Tube (Wampole Laboratories, Cranbury, NJ, USA) was traditionally considered preferable to other culture methods for the detection of brucellae in blood samples [10,11]. Blood specimens are seeded into special vials that contain a mixture of an anticoagulant to prevent clotting and a detergent that disrupts the cellular membranes of white blood cells, releasing phagocyted but still viable microorganisms. The resulting lysate is then centrifuged, and the sediment is dispersed onto appropriate agar plates and incubated. Naturally, the extensive manipulation of the specimen, even if performed in a biological safety cabinet, implies a substantial transmission hazard for the CML personnel.

To avoid exposure, the Centers for Disease Control and Prevention (CDC) have strongly recommended that all laboratory procedures with living brucellae require level 3 biosafety precautions [38]. The organism should be handled in Class II biological safety cabinets by technicians protected by a gown, gloves, goggles, and a respiratory mask [38]. The

drawback of this approach is that, by the time bacterial isolates are suspected or confirmed as *Brucella* species, extensive careless work with the organism has usually occurred, and inadvertent exposure may have already occurred. In 1997, following a large outbreak of LAB in southern Israel, strict infection control practices were rigorously implemented [15]. All blood culture vials flagged positively by the automated Bactec instrument are processed in biological safety cabinets until the possibility of a *Brucella* species is firmly ruled out. Plates are sealed when not in use and properly disposed of and sterilized as soon as the work has been completed, as recommended [30]. Since the antibiotic resistance pattern of the genus is predictable and acquired resistance is uncommon, susceptibility testing of identified *Brucella* organisms has been stopped altogether. Because a prospective comparison of the performance of the Isolator Microbial Tube and the safer automated Bactec system for detecting *Brucella* bacteremia demonstrated a statistically significant advantage of the latter in terms of both sensitivity and time-to-detection [36], the use of the manual lysis-centrifugation system for culturing samples from patients with suspected brucellosis has been utterly discouraged. Since the implementation of this enhanced safety policy, no further cases of LAB have been detected in more than 20 years, despite an ever-growing number of isolations [39]. It seems, then, prudent to recommend in endemic areas that all positive blood culture vials should be initially processed in safety cabinets, pending final identification of the isolate. Since CML technicians in these regions frequently handle *Brucella* organisms, a baseline serological test should be performed upon recruitment and periodically thereafter. This serological monitoring may facilitate the distinction between old and newly acquired infections.

## 6. Brucellar Identification by Traditional Methods

The presumptive identification of members of the genus *Brucella* relies on the typical Gram staining appearance, positive oxidase, catalase, and urease activity, no fermentation of sugars, and lack of motility, and should be confirmed by a molecular method or by a positive slide agglutination reaction with antiserum against the bacterial O-lipopolysaccharide [7]. Each of the individual links of the identification chain is prone to error, misidentifying the isolate and causing LAB. Furthermore, because of the effective veterinarian control of the zoonosis, the disease has become uncommon in industrialized countries, and personnel working at CMLs have become unfamiliar with the phenotypic characteristics of the genus [3]. Gram stain plays an early and key role in correctly identifying *Brucella* species. The presence of small Gram-negative coccobacilli should be the first hint of the true nature of the unknown organism, and no biochemical, MALDI-TOF, or molecular testing should ever be carried out before a thoughtful Gram staining examination of the isolate has been performed. A poor staining technique may result in the classification of brucellae as Gram-positive organisms that can be mistaken for streptococci or corynebacteria [3]. Identifying *Brucella* species by conventional manual methods takes a few days, in the course of which exposure of the laboratory personnel to a highly infectious organism may occur. This is especially hazardous is the catalase test, which is strongly positive in all brucellae and produces bubbling and nebulization of living bacteria [13].

In recent years, commercial systems have gradually simplified these traditional identification methods saving considerable labor time. These kits consist of panels of ready-made dried chemical substrates that, once inoculated with suspensions of the unknown bacterium and incubated, identify the isolate by comparing the test results with those of a comprehensive database. Because of the similarity of the biochemical profiles, these systems do not discriminate between true brucellae and other members of the *Brucellaceae* family and, particularly, the *Ochrobactrum* species (*O. anthropi* [40,41] or *O. intermedium* [40–42]), as well as the taxonomically unrelated *Haemophilus influenzae* [43], *Bergeyella zoohelcum* [44], *Bordetella bronchiseptica* [45], or *Psychrobacter phenylpyruvicus* (formerly *Moraxella phenylpyruvica*) [46]. These unfortunate mistakes have already caused outbreaks of LAB [47], and, therefore, any of these uncommon bacterial species identified by phenotypic methods should be considered a potential *Brucella* organism and, as such, carefully handled in a

safety cabinet until this possibility is firmly excluded. The familiarity of CML personnel with the microbiological features of brucellae, the safe handling of culture media, and the pitfalls in identifying members of the genus should be improved, refreshed, and maintained through periodic education.

## 7. Identification by MALDI-TOF Technology

The recent introduction of MALDI-TOF-based instruments in the CML has profoundly changed how microorganisms are identified. MALDI-TOF technology enables the fast (within minutes), precise, reproducible, and cost-effective identification of bacterial isolates to the species level, substituting the manual, cumbersome, and slow traditional biochemical testing [8,48].

The capability of MALDI-TOF technology to correctly identify brucellae is evolving at a slow pace. Since commercial MALDI-TOF instruments are costly and usually unavailable in developing countries where zoonosis is prevalent, data based on the field evaluation of the method are scarce. Initially, the database reference of the Vitek MS system (bioMérieux, France) misidentified *B. melitensis* as *O. anthropi* [49]. An improved database, named Vitek MS IVD, has been recently added, which includes reference spectra for *Brucella* species, making it possible for the unambiguous discrimination between members of the *Brucella* and *Ochrobactrum* genera, as well as satisfactory speciation of the three most common zoonotic species: *B. melitensis*, *B. abortus*, and *B. suis* [50,51]. The competitor Bruker system (Bruker Daltonics, Germany) did not include the *Brucella* genus protein profile in the original Biotyper reference library, and identification was only possible by employing customized databases [9,52,53]. Nowadays, identification to the genus level is possible using Bruker's Security Relevant Library, which has to be purchased separately and is not available in many countries [8]. The system, however, unreliably discriminates between the different *Brucella* species [9,52]. The latest FDA-approved CDC's MicrobeNet database [54] identifies brucellae to the genus level, while the RUO library identifies them to the species level [55]. However, the use of novel technology for the identification of brucellae, is not risk-free, and the procedures recommended by the manufacturers for other bacterial pathogens are not adequate for the manipulation of biosafety level 3 *Brucella* strains [50]. In a large survey of exposure to the organism among CML personnel of New York City hospitals, inappropriate use of the MALDI-TOF and misidentification of the isolate were responsible for 84% of the events [3]. MALDI-TOF MS analysis should never be applied directly to bacterial colonies growing on agar plates or positive blood culture broth before a Gram stain of the isolate is examined, and other phenotypic features, such as growth conditions and media or colony morphology, are taken into consideration [52]. If small Gram-negative coccobacilli are visualized, there is strict aerobic growth on blood- and chocolate-agar media, and capnophilia and white, non-hemolytic colonies are detected, a *Brucella* species should be strongly suspected. In some CMLs, the broth of positive culture vials and bacterial colonies growing on a plate [50] are directly transferred to the MALDI-TOF matrix without further workup to save time. This practice represents an occupational risk because adding a small volume of solvent/matrix does not completely inactivate live brucellae, probably due to the lack of enough contact between the solvent and the bacteria [3]. To prevent exposure, an initial bacterial inactivation step is mandatory before the protein extraction. The exposure of brucellae to the MALDI-TOF solvent in a tube before it is spotted onto the slide will sterilize the specimen, removing the potential contagion risk [50]. Alternatively, 33% acetonitrile, 33% absolute ethanol, 3% trifluoroacetic acid, 31% water [50], absolute ethanol and formic acid (*v/v* 10%) [9], absolute ethanol, 70% formic acid, and acetonitrile [56,57] can be also successfully employed.

## 8. Identification by Molecular Methods

In recent years, genotypic identification of isolates, instead of the traditional phenotypic methods, is increasingly being employed in CMLs. In most cases, the PCR amplification of the 16S rRNA gene, which is present in all bacteria, is followed by sequencing of the

amplicon. The resulting sequence, which is species-specific, is then compared with those deposited in a database, enabling the precise identification of the isolate. This strategy avoids the pitfalls of conventional biochemical identification and has successfully detected *B. melitensis,* initially misidentified as *O. anthropi* [58]. However, it should be emphasized that molecular identification requires previous isolation of the organism; therefore, biosafety precautions should be followed in handling any aerobic, Gram-negative isolate before subjecting it to molecular identification. The fluorescence in situ hybridization (FISH) assay targeting a segment of the 16S rRNA gene has recently been evaluated by employing simulated blood cultures spiked with different *Brucella* species and biotypes [59]. The novel test can be applied directly to positive blood culture broths enabling the identification of all *Brucella* species pathogenic to humans [59]. The FISH technique is more rapid and cheaper than the sequencing of the 16S rRNA gene and real-time PCR methods for the molecular identification of brucellae and is suitable for resource-limited laboratories. However, because of the low polymorphism of the "universal" 16S rRNA gene sequence among members of the *Brucellaceae* family, the probe used in the assay shows only a single mismatch with the gene of *Ochrobactrum* species, preventing the discrimination between the two genera [59]. To overcome the problem, an unlabeled competitor differing at one base from the probe sequence has been successfully employed, avoiding misidentification of the organism and preventing exposure to dangerous brucellae [59]. Prospective studies employing a variety of clinical specimens are still needed to evaluate the performance of this promising test in regions endemic to zoonosis.

## 9. Post-Exposure Prophylaxis and Other Measures

Following the recognition of the exposure incident, a thorough investigation should be conducted immediately. The event should be reconstructed, documented, and reported to the Public Health authorities; the timing, setting, and circumstances of the exposure event should be determined as precisely as possible, and the results of the investigation should be discussed with the laboratory staff and used for educational purposes and correction of deficiencies. The members of the laboratory personnel potentially involved in the exposure should be identified, and the individual risk should be assessed as high or low following the Centers for Diseases Control and Prevention guidelines [60], as condensed in Table 2.

**Table 2.** Assessment of the exposure risk and indications for post-exposure prophylaxis and monitoring.

| Risk Category | Exposure Setting | | | | | Post Exposure Measures | | |
|---|---|---|---|---|---|---|---|---|
| | Enriched Material [a] and Reproductive Clinical Specimens | | | | Other Clinical Specimens | | | |
| | Work outside of a CCBSC [b] | Work at <5 Feet from Someone Working outside a CCBSC [b] | Work on a CCBSC [b] without PPE [c] | Aerosol-Generating Procedures on an Open Bench | Contact with Mucosae or Broken Skin | Post-Exposure Prophylaxis | Serological Follow-Up | Clinical Monitoring |
| High | Yes | Yes | Yes | Yes | Yes | Yes | 6 months | 6 months |
| Low | No | No | No | No | No | No | 6 months | 6 months |

[a]: Contact with *Brucella* isolates and cultures on solid or liquid media. [b]: Certified Class II biological safety cabinet. [c]: Personal protective equipment (gloves, gown, eyes protection, and mask).

Because of the high infectivity of *Brucella* organisms, the attack rate of clinical disease among exposed laboratory personnel is remarkably high, and 71 LAB cases were diagnosed among 167 exposed workers summarized by Traxler et al. [2]. Therefore, post-exposure prophylaxis consisting of doxycycline (100 mg) orally twice daily and rifampin (600 mg) once daily for a minimum of 21 days should be offered to those considered to be at high risk for LAB and immunosuppressed individuals disregarding the risk level. Trimethoprim-sulfamethoxazole (cotrimoxazole) or another antimicrobial agent effective against *Brucella* should be selected (for at least 21 days) if doxycycline or rifampin are contraindicated, as well as for pregnant women. Serological testing of all the exposed laboratory personnel

should be performed as soon as possible and repeated at 6, 12, 18, and 24 weeks after the last known exposure [3,60], as well as monitoring clinical symptoms and signs, disregarding their risk-assessment classification.

## 10. Conclusions

Brucellosis continues to be a public health problem of worldwide dimensions that poses a substantial risk of transmission to laboratory workers. In regions endemic to the disease, the hazard of contagion of the CML personnel is high, but even in countries where strict control measures are implemented, and zoonosis has been controlled, the accidental transmission of *Brucella* organisms remains a serious concern. Exposure to virulent brucellae may occur at each of the successive steps of the diagnostic chain, from handling clinical specimens through isolation of the organism to its final identification. Although improved culture techniques and novel detection and speciation methods have been added to the CML armamentarium, *Brucella* species still pose a real threat to working personnel. To avoid occupational infections in today's busy and complex laboratory environment, a comprehensive approach is necessary, consisting of educating technicians on the microbiological identification of members of the genus and its pitfalls, strict adherence to safe work practices, and proper use of containment devices and personal protective barriers during the manipulation of unidentified bacterial isolates.

**Funding:** This research received no external funding.

**Institutional Review Board Statement:** Not applicable.

**Informed Consent Statement:** Not applicable.

**Data Availability Statement:** Not applicable.

**Conflicts of Interest:** The author declares no conflict of interest.

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
