# Peer review of "Preventing Laboratory-Acquired Brucellosis in the Era of MALDI-TOF Technology and Molecular Tests: A Narrative Review"

_zoonoticdis, doi:10.3390/zoonoticdis2040016_

Round 1
Reviewer 1 Report
Summary:
The current review article covers topics about accidental brucella infections in lab/work settings, and the prevention and management of such exposures, with a special emphasis on the biosafety of employing the newer detection and identification methods.
Comments:
Page 2: “2. The organism”: The section is not discussing enough about the Brucella organism? “Although each one of these species is associated with preferential animal hosts… million new cases of human infection are annually detected [11]” these sentences cover much about the host.
Page 3: Please avoid “etc”. Give the complete exhaustive information if possible.
Page 3: "flue-like disease” should be “flu-like disease”?
Page 3: “[10, 11 mimicking”. Please check the format.
Page 4: “Whereas in the United States,…”. This paragraph seems to be off, from the lab condition it takes the reader suddenly toward the epidemiology of the disease in different countries/geographical areas. And it is also not very suitable for the section.
Page 4: “[7].
Although” Please check format.
Page 6: “for culturing patients” it may be better to rewrite the sentence as, ‘for culturing samples from patients'.
Page 6: “In recet years”. ‘recent’?
Page 9: Table 2. Please check the font settings.
Author Response
The manuscript entitled “Preventing Laboratory-acquired Brucellosis in the Era of MALDI-TOF Technology and Molecular Tests: a Narrative Review” (zoonoticdis-1910524) has been revised following Reviewer 1 remarks, as follows:
- Page 2: “2. The organism”: The section is not discussing enough about the Brucella organism? “Although each one of these species is associated with preferential animal hosts… million new cases of human infection are annually detected [11]” these sentences cover much about the host.
Reply. The referee is correct. The section discusses the transmissibility of brucellae and not the microbiological features of the genus. The subtitle has been changed to ”Brucella: a Highly Transmissible Organism” to better reflect its content.
- Page 3: Please avoid “etc”. Give the complete exhaustive information if possible.
Reply. I believe the reviewer refers to Table 1. “etc” has been deleted and additional information has been added in the revised version of the manuscript (“and malfunction of biological safety cabinets”).
- Page 3: "flue-like disease” should be “flu-like disease”? Reply.
- The error has been corrected in the revised version.
- Page 3: “[10, 11 mimicking”. Please check the format.
Reply. The sentence and format have been modified, as indicated by the referee as follows: “The disease may affect different body organs such as the joints and bones, the liver, the genital tract, and the central nervous system,[10, 11] mimicking other infectious and non-infectious conditions…”.
- Page 4: “Whereas in the United States,…”. This paragraph seems to be off, from the lab condition it takes the reader suddenly toward the epidemiology of the disease in different countries/geographical areas. And it is also not very suitable for the section.
Reply. The paragraph has been modified as follows: “The burden of the disease in endemic areas can be appalling, posing a permanent threat to laboratory personnel. Whereas in the United States, approximately 120 cases of brucellosis are reported annually countrywide and LAB events are rare [4], in a single CML in Ankara, Turkey, a mean of 400 clinical specimens yield Brucella organisms each year, and LAB had affected 10 of 55 (18%) technicians, with an annual risk of 8% per employee [19]”.
- Page 4: “[7].
Although” Please check format.
- I’m sorry, but I could not understand the reviewer’s remark.
- Page 6: “for culturing patients” it may be better to rewrite the sentence as, ‘for culturing samples from patients'.
Reply. The sentence has been modified as suggested by the reviewer.
- Page 6: “In recet years”. ‘recent’?
Reply. The misspelled word has been corrected.
- Page 9: Table 2. Please check the font settings.
Reply. The font and spacing have been modified.
In addition, the manuscript has been revised by a native English speaker.

Reviewer 2 Report
I found this mansucript "Preventing LAB in the era of MALDI-TOF..." by Pablo Yagupsky to be a thorough and needed addition to the body of work related to handling and diagnostic measures for the often underappreciated pathogen, Brucella spp. The author was thorough in including relevant literature, case reports, and other information related to outbreaks and brought that information into the framework of diagnostics. I have no major comments. I support the publication of this important manuscript.
Author Response
Thank you very much for your suggestion.